# Wavelet Transforms Significantly Sparsify and Compress Tactile Interactions

**DOI:** 10.3390/s24134243

**Published:** 2024-06-29

**Authors:** Ariel Slepyan, Michael Zakariaie, Trac Tran, Nitish Thakor

**Affiliations:** 1Electrical and Computer Engineering Department, The Johns Hopkins University, Baltimore, MD 21218, USA; trac@jhu.edu; 2Biomedical Engineering Department, The Johns Hopkins University, Baltimore, MD 21218, USA; mzakari4@jhu.edu

**Keywords:** compression, prosthetics, sparsity, spatiotemporal, tactile sensing, wavelet transform

## Abstract

As higher spatiotemporal resolution tactile sensing systems are being developed for prosthetics, wearables, and other biomedical applications, they demand faster sampling rates and generate larger data streams. Sparsifying transformations can alleviate these requirements by enabling compressive sampling and efficient data storage through compression. However, research on the best sparsifying transforms for tactile interactions is lagging. In this work we construct a library of orthogonal and biorthogonal wavelet transforms as sparsifying transforms for tactile interactions and compare their tradeoffs in compression and sparsity. We tested the sparsifying transforms on a publicly available high-density tactile object grasping dataset (548 sensor tactile glove, grasping 26 objects). In addition, we investigated which dimension wavelet transform—1D, 2D, or 3D—would best compress these tactile interactions. Our results show that wavelet transforms are highly efficient at compressing tactile data and can lead to very sparse and compact tactile representations. Additionally, our results show that 1D transforms achieve the sparsest representations, followed by 3D, and lastly 2D. Overall, the best wavelet for coarse approximation is Symlets 4 evaluated temporally which can sparsify to 0.5% sparsity and compress 10-bit tactile data to an average of 0.04 bits per pixel. Future studies can leverage the results of this paper to assist in the compressive sampling of large tactile arrays and free up computational resources for real-time processing on computationally constrained mobile platforms like neuroprosthetics.

## 1. Introduction

Prosthetics, wearables, and other biomedical sensors utilize high-density sensor arrays for touch [1], grasp [2], texture palpation [3], and object recognition [4]. Of particular emergent interest is dexterous robotic and prosthetic hands equipped with tactile sensor arrays. As the spatiotemporal resolution of tactile sensor arrays in robotics increases, new sampling and data management techniques are needed to support the higher throughputs of high-density spatial and temporal sampling (Figure 1).

To circumvent the sampling requirements, a recent trend is using compressive sampling to take fewer measurements and reconstruct the pressure profile based on sparsity [5,6]. Yet, an investigation into the basis of sparsity to use for the compressive sampling of tactile interactions is still lacking, and accurate reconstruction depends on representing the data in its sparse domain [7]. Previous research on compressive tactile sensing has mainly used the discrete cosine transform (DCT); however, tactile interactions are known to be spatiotemporal, and the DCT may not be the best sparsifying transform [8,9].

From the data management perspective, applying sparsifying transformations to tactile data can greatly improve efficient data storage and enable real-time processing through compression [10,11]. The data management problem is especially critical for high spatiotemporal resolution (HSR) sensor arrays as the increased spatiotemporal resolution can generate massive data streams. For example, some HSR tactile sensor arrays like the kilohertz–kilotaxel tactile sensor array (4096 taxels sampled at 5 k Hz) already generate 10 mb/s of tactile data [12]. Big data streams like this can massively constrain real-time processing methods used in closed-loop control, especially for computationally limited mobile platforms such as neuroprosthetics and robotics.

Therefore, from both a sensing and data management perspective, it is necessary to identify the sparsifying transform for HSR tactile sensor arrays. A promising class of sparsifying transforms are wavelet transforms. Unlike Fourier transforms, which provide frequency information globally, wavelet transforms offer a joint time-frequency representation calculated using specific filter functions which can be formulated in a wide variety of shapes or patterns [13]. This allows for pinpointing specific frequencies within a signal and their corresponding time of occurrence. This characteristic is particularly well-suited for tactile data because tactile interactions are known to be spatiotemporally sparse and active regions are often localized in both space and time [14]: for example, in any given tactile palpation only a few pixels may be used and active at any given time. As this paper will demonstrate, wavelet transforms can provide an effective sparse representation of tactile interactions. Moreover, this sparse representation is often sparser than what can be obtained by using traditional techniques like DCT which operate on the entire signal and may not capture the localized nature of tactile data. Additionally, wavelet transforms can be efficiently implemented in hardware using lifting schemes, making real-time processing practical [15].

Thus, the aim of this work is to develop and test sparsifying transforms, and their best parameters, for HSR tactile interactions to identify which wavelet transformations are best fitted for these high-density tactile sensor arrays. In this paper, we limit the scope of our approach to only discrete wavelet transforms, specifically orthogonal and biorthogonal types which are energy preserving and have a linear phase, respectively. We test the sparsifying transformations in a publicly available high-density tactile object grasping dataset (a 548-sensor tactile glove) and compare the tradeoffs of different transforms in compression, sparsity, and energy retainment. Finally, we investigate which dimension wavelet transform (1D, 2D, or 3D) would best compress the tactile interactions. In this case, the 1D transform would be applied to each taxel independently over time capturing only temporal information, the 2D transform would be applied to each ‘tactile frame’ capturing only spatial information, and the 3D transform would be applied to the volume of tactile data in space and time, capturing spatiotemporal information. An overview of the wavelet-transform based sparsification methodology is shown in Figure 2.

## 2. Methods

### 2.1. High-Density Tactile Dataset

We used a publicly available high-density tactile grasping dataset obtained with the scalable tactile glove (STAG), which has 548 sensors at a density of 16 sensors/cm ^2^ [2]. The grasping dataset contains 26 different grasps of different objects, representing a wide range of relevant daily activity. These grasps include daily objects such as a mug, pen, scissors, ball, and spoon, among others. The objects were grasped for about 5 s numerous times, totaling 33,550 tactile data frames in the dataset. In this study, this dataset was denoised using a second-order infinite impulse response low-pass filter at a pass-band frequency of fs/7.

### 2.2. Discrete Wavelet Transform

For the discrete wavelet transform (DWT), we tested the orthogonal and biorthogonal wavelets available in the MATLAB Wavelet toolbox version 6.2 (R2022b) [16]. Orthogonal wavelets were chosen due to their unbiased denoising abilities [17]. Biorthogonal wavelets were chosen because they have a linear phase which reduces the distortions caused by compression [18]. The level of the wavelet transform was chosen as the maximum decomposition level: N = fix(log2(L)), where L is the length of the input signal, fix rounds down to the nearest integer, and N is the decomposition level. For 2D and 3D, the same rule was followed, where L is the length of the smallest dimension. The DWTs were all implemented in MATLAB R2022b using the wavedec, wavedec2, and wavedec3 functions. The implemented DWT algorithm operates by performing cascading convolutions and decimation. The input signal is convolved with a scaling filter LoD and a wavelet filter HiD, the output of each filter is downsampled, and the low-pass branch is iterated on until the level of transform is reached (Figure 3). The level of the wavelet transform was chosen as the maximum decomposition level: N = fix(log2(L)), where L is the length of the input signal, fix rounds down to the nearest integer, and N is the decomposition level. Each filter stage captures a different frequency content of the signal due to the downsampling. This produces a time–frequency representation of the input signal. After compression, signals are reconstructed using the corresponding waverec, waverec2, and waverec3 MATLAB functions using the matching synthesis filters and upsampling.

We tested 75 wavelets from 12 different families (Figure 4): Daubechies (db1–db10), Coiflets (coif1–coif5), Symlets (sym2–sym10), Fejér-Korovkin (fk4, fk6, fk8, fk14, fk18, fk22), Best-localized Daubechies (bl7, bl9, bl10), Morris minimum-bandwidth (mb4.2, mb8.2, mb8.3, mb8.4), Beylkin, Vaidyanathan, Han linear-phase moments (han2.3, han3.3, han4.5, han5.5), Discrete Meyer, BiorSplines, and ReverseBior (bior/rbio 1.3, 1.5, 2.2, 2.4, 2.6, 2.8, 3.1, 3.3, 3.5, 3.7, 3.9, 4.4, 5.5, 6.8). These wavelets have a broad range of symmetry, support length, and number of vanishing moments. These properties affect how well suited they are for tactile data. For example, high symmetry will provide less phase distortion, which is important for accurate signal reconstruction [19]. A large number of vanishing moments usually results in more wavelet coefficients being zero, which is beneficial for data compression by reducing the amount of information that needs to be stored or transmitted [19]. Support length determines the extent of the signal that influences each coefficient; shorter support lengths can lead to better localization in time but might compromise frequency resolution [19]. Therefore, choosing the appropriate wavelet involves balancing these factors to optimize the representation and processing of tactile sensor data.

We implemented the sparsification protocol as outlined in Figure 2. We varied reconstruction error after sparsification and recorded three parameters that evaluated the candidate sparsification process: sparsity, number of bits per pixel (NBP), and energy ratio (ER). Sparsity was defined as the percentage of non-zero values compared to the initial length. The number of bits per pixel was defined as the number of bits necessary to represent all the non-zero values. Energy-ratio was the retained Euclidean energy of the sparsified data compared to its un-sparsified version.

To sparsify the data after wavelet transformation, we used quantization: (1)$$sparse\_rep=Q\cdot fix\;(\frac{wavelet\_rep}{Q})$$
where Q is the quantization factor, and fix rounds the nearest integer towards 0. Quantization forces the wavelet coefficients to be integer multiples of Q (Figure 5). Thus, our search process consisted of adjusting the sparsifying Q factor for each wavelet transformation until a desirable reconstruction error (normalized mean square error, NMSE) was reached. Then, the sparsity, number of bits per pixel, and energy ratio were recorded for that trial.

To find the Q factor that achieved a particular NMSE we conducted a binary search, augmenting Q until the appropriate value was reached. We tested the wavelets in 4 different NMSE values: 0.01, 0.0043, 0.0015, and 0.0001. The NMSE is calculated as follows:(2)$$NMSE=\frac{1}{n}\sum_{i=1}^{n}\frac{{\left(y_{i}-{\hat{y}}_{i}\right)}^{2}}{\bar{y}}$$
where y is the original signal, $\bar{y}$ is the average of the original signal, and $\hat{y}$ is the reconstructed signal after sparsification. This method is summarized in Algorithm 1.
**Algorithm 1.** **Sparsifying Survey****Input:** sensor data **D**, candidate wavelet transform **W**, desired **NMSE****Output: sparsity**, **NBP**, **ER, x_recon**1:  Apply transform **W** to **D** → obtain **wavelet_rep**2:  **while** NMSE is not reached3:      **sparse_rep** = **Q** * *fix*(**wavelet_rep/Q**)4:      **x_recon =**
*waverec*(**sparse_rep**) //recover x5:      Calculate **NMSE** between **x_recon** and **D**6:      Check if **NMSE** is too big or too small7:      Adjust **Q** using binary search method8:  **end**9:  Calculate **sparsity** of **sparse_rep**10: Calculate average number of bits per pixel (**NBP**) of **x_recon**11: Calculate energy ratio (**ER)** between **sparse_rep** and **wavelet_rep**

## 3. Results

In this section we present extensive results on the applications of wavelet transformation to tactile data; we provide the following:(1)Tabular Summary of Results. We present key measurements from our testing, namely the top performing wavelets in sparsity, compression, and energy-retainment sense for each dimensional transform tested (1D/2D/3D). Additionally, 1D/2D/3D DCT is presented as a comparison in the table.(2)Compactness of Wavelet Representation. We present the efficiency of wavelet representations of tactile data by investigating the importance of a few, large-magnitude wavelet coefficients on producing accurate reconstructions. We also present the ranking order of these coefficients and the compactness among the top-performing transforms.(3)Sparsity of Wavelet Representation. We present the sparsity of each candidate wavelet transform tested and highlight the effect of dimensionality on average. Additionally, we show aggregate results across different NMSE values and highlight the top performers in each dimension.(4)Effect of Dimensionality on Spatiotemporal Error. We highlight the differences in spatial and temporal errors that result when evaluating the wavelet transform temporally (1D), spatially (2D), and spatiotemporally (3D).(5)Spatiotemporal Reconstruction of Tactile Data. Building on the effect of dimensionality, we present a reconstruction of tactile data using the compressed wavelet representations after evaluating the different dimensionality transforms. Temporal and spatial plots are shown emphasizing the different effects of each transform dimension.(6)Effect of Filter Size on Reconstruction Error and Sparsity. We present the evolving dependence of filter size on reconstruction error and sparsity. We highlight how the size of the sparsest wavelets changes depending on the desired reconstruction error.(7)Similarity of Scaling Functions to Tactile Data. Lastly, we present the similarity between the best performing wavelets and the grand-average tactile interaction. Furthermore, we show how similarity evolves for different reconstruction fidelities.

Tabular Summary of Results. Table 1 shows a summary of key measurements from our testing. The results are grouped by the dimensionality of the transform, and each dimensional transform shows its top five candidate wavelet transforms for each metric (sparsity, average bits per pixel, and energy ratio). Conditional color formatting is used to show ‘good’ values as green and ‘bad’ values as red across all dimensions, and the NMSE is set to 0.01 across all candidates. A comparison with the DCT is shown for each dimensional transform. In general, nearly symmetric or symmetric wavelets such as db2 and sym4 are most sparsifying and compressing for 1D and 3D. For 2D, the haar wavelet is the most sparsifying and compressing (db1/bior1.1/rbio1.1). For energy conservation in all dimensionality cases, the wavelets that are incoherent to tactile data, such as bior3.1, are optimal. The full survey results are provided at the end of the paper (in the Data Availability Statement section).

Compactness of Wavelet Representation. The sparsest performing candidates in each wavelet family were selected and reconstructed over a short exemplary segment of the data with varying numbers of coefficients (Figure 6). The NMSE and PSNR were recorded for each number of coefficients, creating phase-transition curves. A sorted list of coefficient magnitudes for each transform is also shown. The left-most curves show more efficient representations, with only a few large coefficients being necessary to achieve low NMSE error and high PSNR. The most efficient candidate transforms in each family all have short supports such as db2 and fk4.

Sparsity of Wavelet Representation. Figure 7 shows a plot of sparsity for each candidate wavelet transform at an NMSE of 0.01, with each candidate transform represented as a point with a particular shape, face color, and edge color. This is performed to show each candidate wavelet transform encoded as a unique point. Points are distributed horizontally for easier viewing, and horizontal displacement is arbitrary. The underlying-colored regions of red, green, and blue show the sparsity distribution of each dimensional transform. The center of each colored region is placed at the average sparsity for sparsifying transforms of that dimension, and the span of the region shows the standard deviation of the sparsity. The red, green, and blue regions represent the sparsity distributions of 1D, 2D, and 3D transforms, respectively. For each NMSE, 1D transforms are the most sparsifying, followed by 3D and then 2D. Figure 8 shows the sparsity distribution as it evolves over the different tested NMSEs. In all cases, the sparsity decreases as the desired reconstruction error decreases. Additionally, the evolution of sparsity and compression rate (bits per pixel) for reconstruction error is highlighted for the sparsest candidate in each dimensionality case.

Effect of Dimensionality on Spatiotemporal Error. The quantitative effect of transform dimensionality is shown in Figure 9. To assess the bias of the choice of dimensionality, the NMSE of each reconstruction is recalculated temporally and spatially using the sparsest wavelet in each transform group. For the temporal recalculation, the NMSE of each sensor over time is calculated independently. For the spatial recalculation, the NMSE of each spatial frame is calculated independently for every time point. In all cases, the average NMSE is 0.01. Dividing the calculation shows how the variance of temporal and spatial error depends on the dimensionality of the transform. Temporally, the 1D transform has the least varying error across all sensors (STD = 0.004). The 2D and 3D transforms have larger standard deviations in error (0.006 and 0.007, respectively). Spatially the 1D transform has the largest standard deviation (STD = 0.006), the 2D transform has the lowest standard deviation (STD = 0.003), and the 3D transform is in between (STD = 0.004).

Spatiotemporal Reconstruction of Tactile Data. To visually assess the effect of dimensionality, the sparsest candidate reconstructions from each dimensional transform (all with the same NMSE of 0.01) were plotted over time. Each curve represents the reconstruction of a sensor in the array (Figure 10). The same frame during the start of the grasp is shown for the three cases to highlight the differences in reconstruction between 1D, 2D, and 3D. Visually, the 1D reconstruction has the closest similarity for the high magnitude sensors over time, but with sporadic errors when plotted spatially. The opposite is true for the 2D case, where high magnitude sensors have close similarity spatially, but sporadic errors occur when plotted temporally. The 3D case lies in the middle between 1D and 2D. The frame error of each reconstructed frame is emphasized to help discriminate the spatial differences.

Effect of Filter Size on Reconstruction Error and Sparsity. The relationship between filter size and sparsity for each evaluated NSME threshold is shown for 1D transforms in Figure 11. For 2D and 3D transforms, the filter size and sparsity were always related, regardless of NMSE thresholds, with smaller filters generally leading to sparser representations. For 1D transformations, high-fidelity reconstructions are obtained from sparser representations when filter sizes are larger. For low-fidelity approximations, sparser representations are obtained with smaller filter sizes. The trend pivots between these two extremes, where middle fidelity reconstructions have no relationship between filter size and sparsity. As reconstruction error decreases, larger wavelets produce sparser representations. This trend is shown with a line of best fit with corresponding 95% confidence interval markings.

Similarity of Scaling Functions to Tactile Data. The sparsest wavelet candidates for each NMSE threshold are shown in Figure 12, along with the grand average temporal tactile interaction.This grand average tactile interaction is obtained by aligning all the recordings of tactile grasping in the dataset and averaging the trials. Notably, the best-fit candidate wavelets share strong visual similarity to the grand average interaction, specifically with the scaling functions.

In summary, the primary results from this study are as follows:(1)Wavelet transforms produce compact and highly sparse representations (up to 0.5%) of tactile interactions with high compressibility (average of 0.04 bits per pixel). The Symlets 4 wavelet applied in 1D produces the sparsest representation of tactile data in coarse approximations, and Biorthogonal 6.8 produces the sparsest representations for high-accuracy reconstructions.(2)Evaluating the wavelet transform temporally (1D) produces much sparser representations than evaluating the wavelet transform spatially (2D); and evaluating the wavelet transform spatiotemporally (3D) lies in between. However, when focusing on temporal or spatial representations, the other will have sporadic occurring errors.(3)The size of the 1D wavelet filter impacts the reconstruction error, and larger wavelets produce sparser representations when high accuracy is desired. Conversely, to yield approximate reconstructions, shorter wavelet filters produce sparser representations.(4)Highly sparsifying wavelets share visual similarity to temporal tactile data.

## 4. Discussion

The numerical results suggest that wavelet transforms are excellent candidates for sparsifying tactile data. When evaluated temporally, they can yield sparsification rates of 0.5% and compress tactile data to an average of 0.04 bits per pixel with a minimal reconstruction NMSE of 0.01 (Table 1). Furthermore, the 1D DWT yields up to 3.4 times sparser representations than the ‘conventional’ sparsifying DCT. However, the 1D DCT is more sparsifying than 2D DWT and 3D DWT. The representations that the wavelet transforms produce can be highly compact, with highly sparsifying wavelets showing phase-transition points at ~10% with a few large magnitude values being important for accurate reconstruction and many small coefficients being insignificant (Figure 6).

From the investigation of dimensionality, we see that temporal sparsification (1D) yields the sparsest representations, 2D spatial transforms yield the densest representations, and 3D spatiotemporal transforms lie in between (Figure 7 and Figure 8). However, as expected, 2D transforms preserve spatial structure with less deviation in error and 1D transforms preserve temporal structure but have significant deviations in spatial error (Figure 9). This is expected because 1D and 2D transforms are better suited for capturing temporal 1D data and spatial 2D data, respectively.

Our investigation on the relationship between filter size, sparsity, and reconstruction error showed changing trends for 1D transformations. Interestingly, we found that to achieve high-fidelity reconstruction, wavelets with large supports like Biorthogonal 6.8 produced the sparsest representation, and the opposite was true for coarse approximations (Figure 11). This suggests that tactile data have significant low-frequency content when evaluated temporally, and that large smooth filters are better at approximating tactile interactions with high accuracy.

Lastly, as expected we saw that the optimal sparsifying wavelets share a significantly similar shape to actual tactile data. By plotting the grand average temporal tactile interaction, we saw substantial similarity to the scaling functions of the best wavelets (Figure 12). This reinforces a predominance of smooth, low-frequency content in tactile interactions.

The dataset we used contained 33,550 tactile frames from a 548-sensor tactile glove collected during the grasping of 26 different objects used in daily life. While this is a large and diverse dataset, the results of this survey should be implemented with caution when transitioning to new datasets; however, general trends are likely to persist, and future studies can utilize the pipeline and open-source code presented in this paper to test additional datasets.

## 5. Conclusions

In conclusion, the findings of this study underscore the effectiveness of wavelet transforms, particularly the 1D discrete wavelet transform (DWT), in sparsifying tactile data. The temporal evaluation revealed impressive sparsification rates of 0.5%, compressing tactile data to an average of 0.04 bits per pixel with a minimal reconstruction NMSE of 0.01. Comparisons with other transforms indicated the superiority of 1D DWT in producing sparser representations. Additionally, the investigation into dimensionality highlighted that temporal (1D) sparsification yielded the sparsest representations, while 2D spatial transforms retained spatial structure with less error deviation. The relationship between filter size and sparsity revealed nuanced trends, emphasizing the advantage of large smooth wavelets, such as Biorthogonal 6.8, for high-fidelity reconstruction; and smaller wavelets such as Symlets 4 for coarser approximations. The alignment of optimal sparsifying wavelets with the shape of actual tactile data supported the prevalence of smooth, low-frequency content in tactile interactions.

Wavelet transforms are excellent candidates for sparsifying tactile interactions. They can achieve high sparsification and high compression while retaining low reconstruction errors. Following the results of this work, future devices can leverage the extreme temporal sparsity of tactile interactions to improve compressive sampling or data compression. Additionally, implementing these sparsification techniques in tactile sensor arrays may reduce sampling requirements and power consumption while improving noise tolerance—freeing computational resources for mobile platforms like robotics and neuroprosthetics.

## Figures and Tables

**Figure 1 sensors-24-04243-f001:**
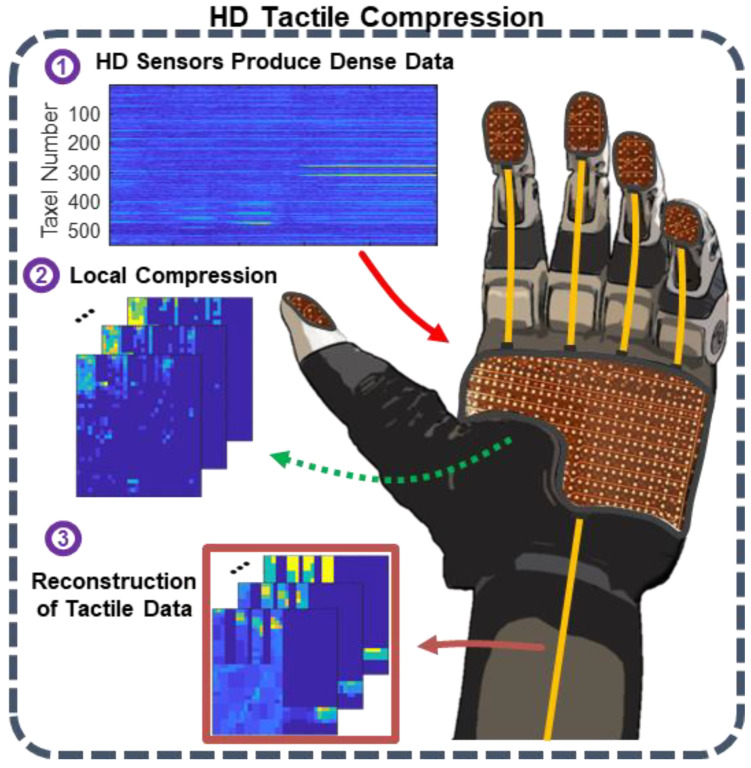
High-density tactile compression in a robotic hand. (1) High-density tactile sensors produce dense spatiotemporal tactile data. (2) Local compression is employed to significantly reduce the bandwidth for data transmission of the robotic hand. (3) Tactile data are reconstructed from the measured compressed outputs.

**Figure 2 sensors-24-04243-f002:**
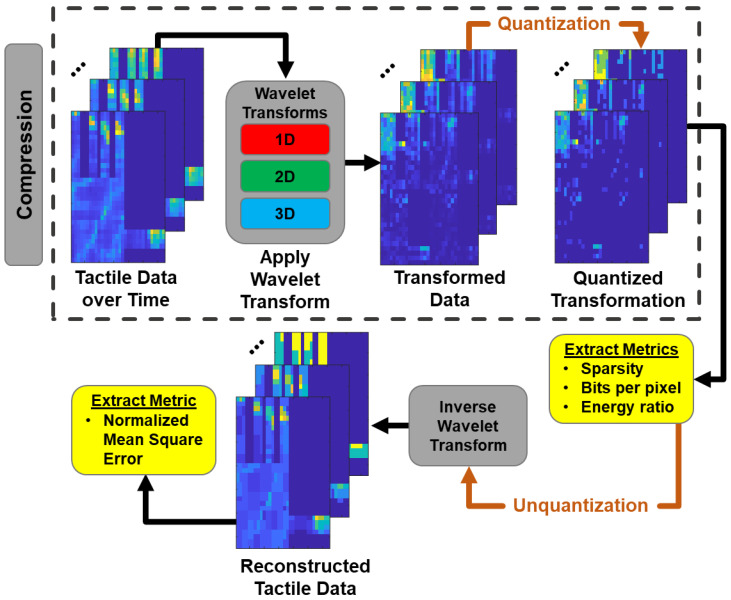
Overview of our wavelet transform-based sparsification approach applied to tactile compression scheme in the 2D case. The sparsification scheme is laid out as follows: tactile data are collected from the sensor array and a wavelet transform is performed on each collected frame. The values after transformation are quantized and can now be transmitted or stored. To recover the compressed data, they must be unquantized and the inverse wavelet transform should be applied. The reconstructed data will have small deviations from the original data due to losses during quantization. During the survey, metrics such as the sparsity, average number of bits per pixel, energy ratio, and normalized mean square error are calculated. The tactile sensor array presented has 548 sensors arranged in a 32 × 32 matrix, and lighter colors represent higher pressures.

**Figure 3 sensors-24-04243-f003:**
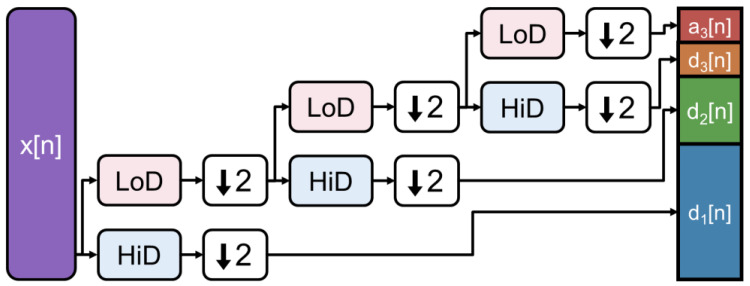
Three-level discrete wavelet transform implemented as cascading filterbanks with downsampling. LoD represents the low-pass filter and HiD represents the high-pass filter. ↓2 represents downsampling by 2. x[n] is the input signal. ai[n] is the approximation coefficients at level i. di[n] are the detail coefficients at level i. The output length is equivalent to the input length.

**Figure 4 sensors-24-04243-f004:**
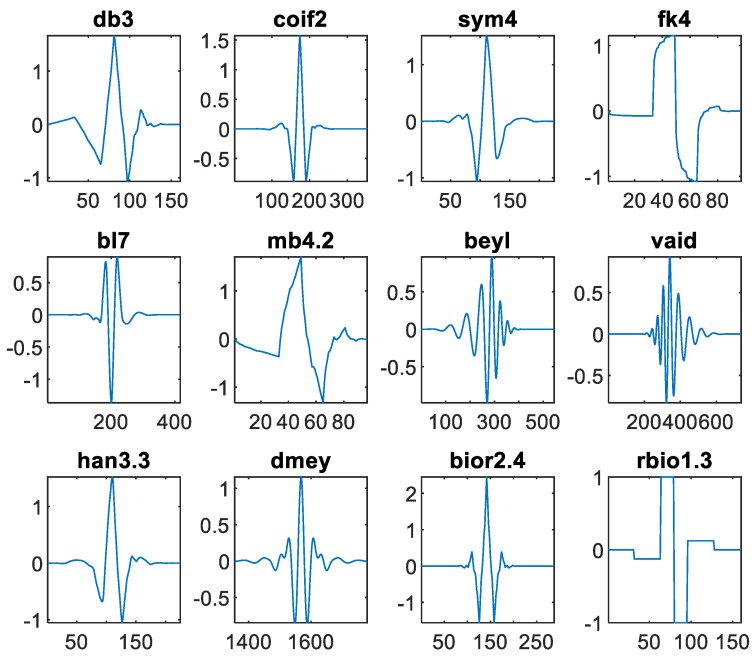
Twelve wavelet families used in this survey. An example of the wavelet function is shown for each. The number of iterations used is 5. Discrete Meyer (dmey) is zoomed in on the x-axis for clarity.

**Figure 5 sensors-24-04243-f005:**
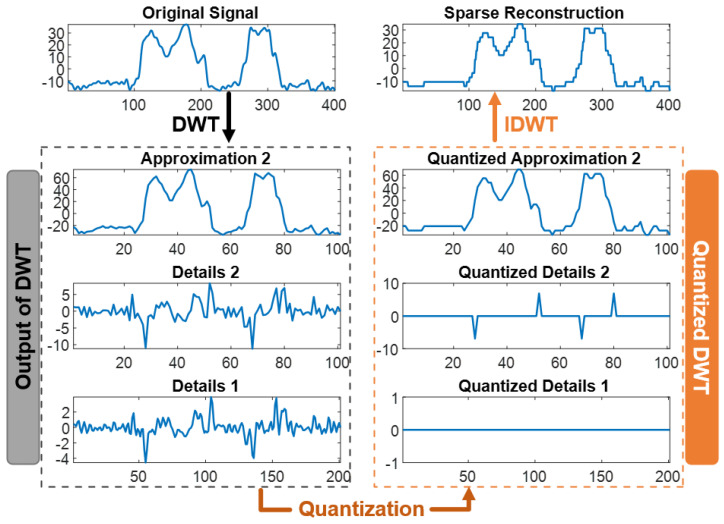
Sparsification and compression by quantization. Quantization is performed with Equation (1) to achieve a desired NMSE = 0.01. The left side shows the original signal and its unquantized wavelet representations. The right side shows the reconstructed signal and the quantized wavelet representations. Small magnitude coefficients are quantized to zero, generating sparsity in the representation.

**Figure 6 sensors-24-04243-f006:**
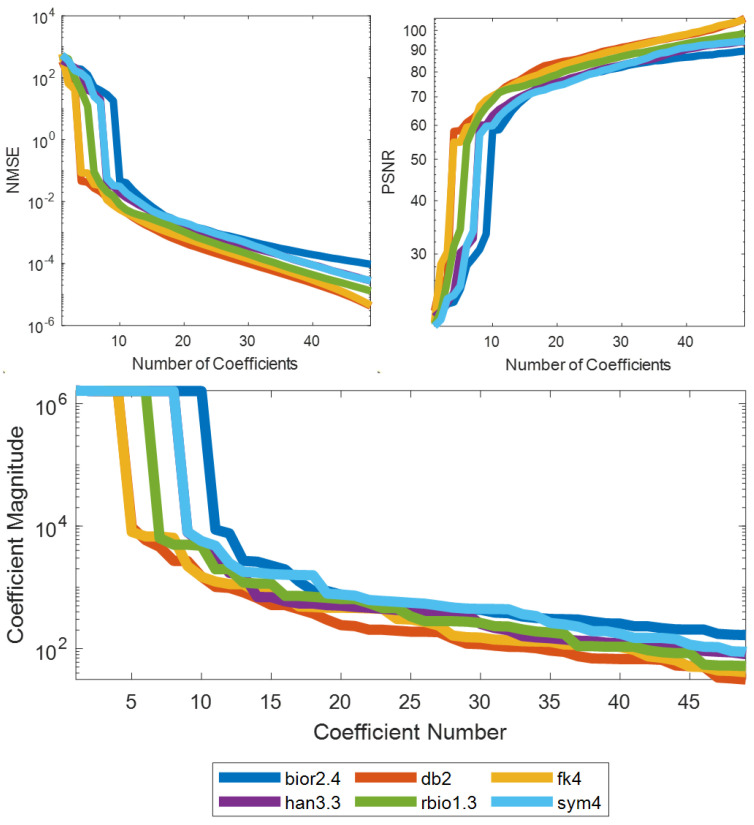
Compactness of 1D wavelet representation of tactile data of an exemplary grasping event. The effect of number of coefficients on NMSE and PSNR, and their ranking order for an exemplary grasping event is shown. A legend is shown at the bottom of the figure to associate the lines in the plots with their matching transform. The best-performing 1D wavelet transforms from a few families are highlighted.

**Figure 7 sensors-24-04243-f007:**
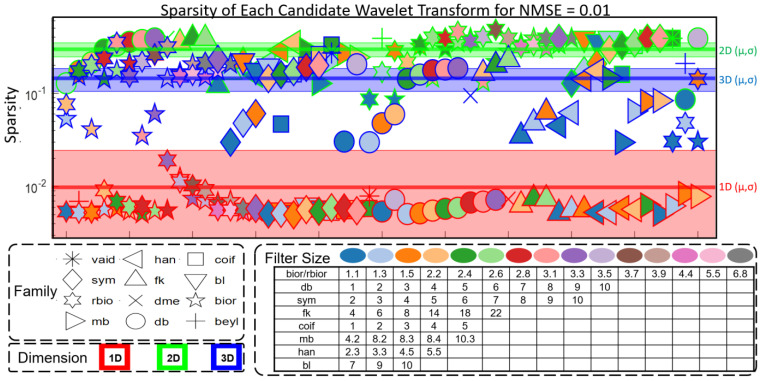
Sparsity of candidate transforms and dimensionality trend. Each candidate transform is represented as a marker with a particular shape, edge color, and face color that encodes its name and size using the legend shown at the bottom of the figure. Colored regions represent the distribution of sparsity for each dimensional transform. Red, green, and blue represent 1D, 2D, and 3D transforms, respectively. Each region is centered at the mean value (darker line) and its vertical span shows the standard deviation of sparsity for that dimensional transform.

**Figure 8 sensors-24-04243-f008:**
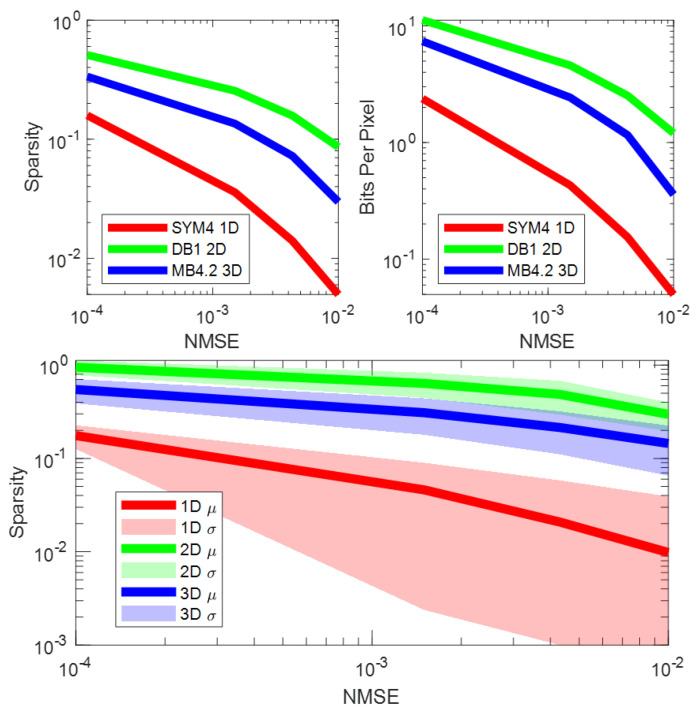
Sparsity and compressibility of the best candidate wavelet transforms for each dimensionality case versus NMSE in the left. The sparsity of each dimensionality case in aggregate is presented in the right.

**Figure 9 sensors-24-04243-f009:**
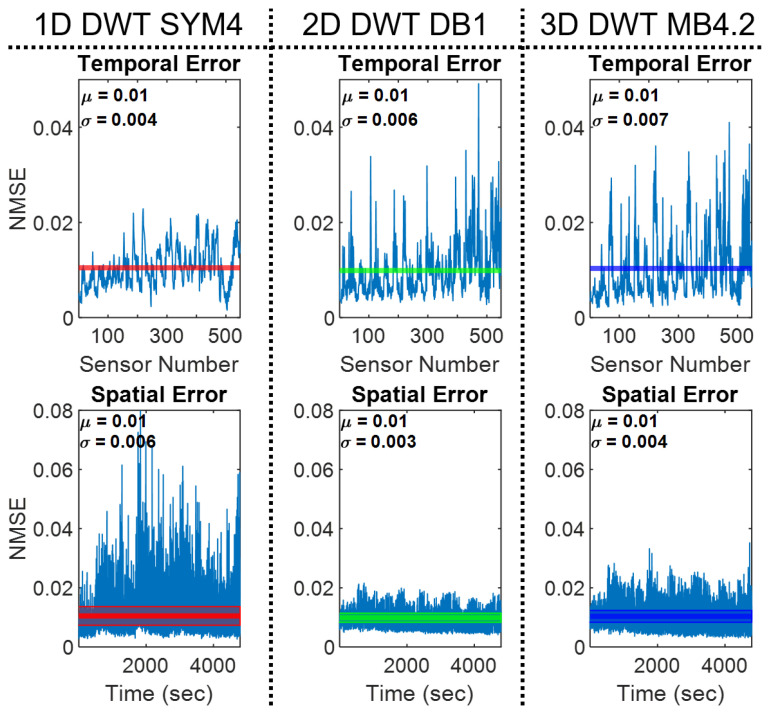
Temporal and spatial errors of tactile reconstruction for the sparsest wavelet transforms in each dimensionality group. The temporal error is calculated as the error over time for each sensor independently and the spatial error is calculated as the error at each spatial frame for each time point independently. All 3 reconstructions have the same average NMSE error (0.01), but different spatial and temporal standard deviations depending on the dimensionality of the transform used. Solid lines represent the average error in each graph, and colored zones represent the span of the standard deviation.

**Figure 10 sensors-24-04243-f010:**
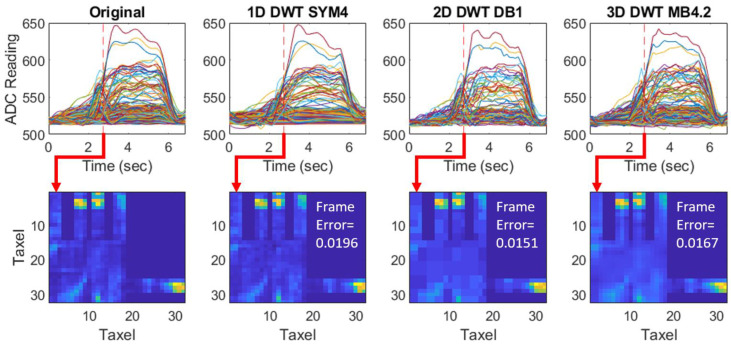
Visual effect of transform dimensionality on tactile reconstruction of a single selected grasp. Top row shows the tactile data over time for each sensor in the array during the grasp. The original curves are shown on the left, and the reconstructed curves for each dimensional transform are shown in the following graphs with each having an overall equivalent NMSE of 0.01. The red dashed line in each graph shows the selected frame during the grasp for which the pressure over the array is plotted spatially. The frame error of each reconstructed frame is emphasized to help discriminate the spatial differences. Respectively, 1D has the highest spatial error of 0.0196, 2D has the lowest spatial error of 0.0151, and 3D has the middle spatial error of 0.0167.

**Figure 11 sensors-24-04243-f011:**
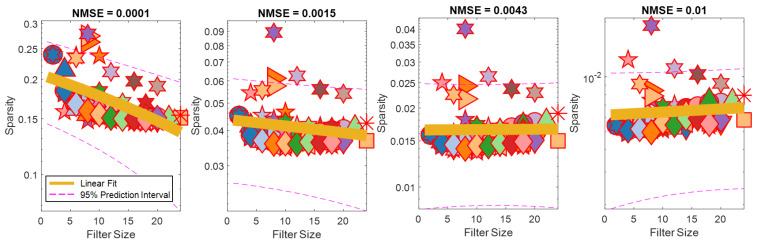
Relationship between filter size and sparsity for different NMSE thresholds for 1D transforms. Markers are coded using the same legend as Figure 7. The trend line is shown in yellow as a linear fit of the data, with a dashed pink line showing the associated 95% confidence interval. Relationship between filter size and sparsity changes from smaller filters being most sparsifying for coarse estimates (NMSE = 0.01), to having no relationship at NMSE = 0.0043, to having larger filters be more sparsifying for high-accuracy reconstructions (NMSE = 0.0001).

**Figure 12 sensors-24-04243-f012:**
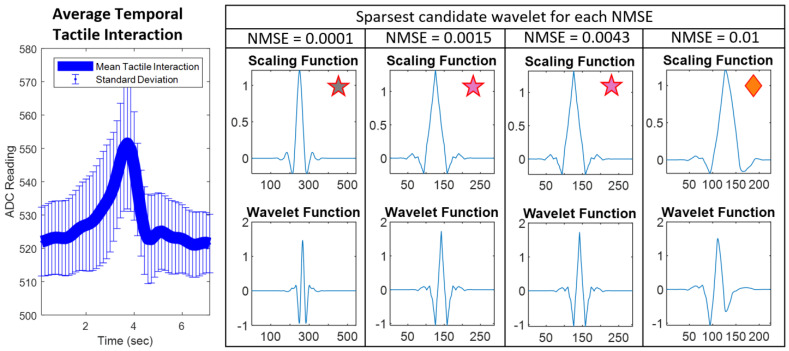
The most sparsifying candidate wavelets for each NMSE and their similarity to the grand average tactile interaction. The grand average tactile interaction is shown on the left, with labeled standard deviation. The scaling and wavelet function of the sparsest candidate for each NMSE value are plotted below the respective section. They are labelled with their corresponding color-coded markers for comparison to other figures. These best wavelets are all in the 1D case: Bior6.8, Bior4.4, Bior4.4, and Sym4 for NMSEs of 0.0001, 0.0015, 0.0043, and 0.01, respectively.

**Table 1 sensors-24-04243-t001:** Summary of results for NMSE = 0.01. Full tabular results for all NMSEs and all data are shown at the end of the paper (in the Data Availability Statement section). Conditional color formatting is used to show high sparsity, low average bits per pixel, and high energy ratio as green and with the low sparsity, high bits per pixel, and low energy ratio as red, across all dimensions. The ranking order refers to sparsity with 1 being the sparsest. The 1D/2D/3D DCT transform is shown as well for comparison as a default sparsifying transform.

Dimension DWT	Ranking	Sparsity		Bits per Pixel		Energy Ratio	
		Value	Name	Value	Name	Value	Name
**1D DWT**	1	0.00497	‘sym4’	0.040618	‘db2’	1	‘bior3.1’
	2	0.00508	‘db2’	0.040618	‘sym2’	0.985	‘rbio3.1’
	3	0.00508	‘mb4.2’	0.040619	‘mb4.2’	0.971	‘bior3.3’
	4	0.00508	‘sym2’	0.043318	‘bior1.1’	0.97	‘dmey’
	5	0.00511	‘bior2.4’	0.043318	‘db1’	0.962	‘vaid’
**1D DCT**		**0.01693**	**‘DCT’**	**0.135513**	**‘DCT’**	**0.7985**	**‘DCT’**
**2D DWT**	1	0.086048	‘db1’	1.204673	‘db1’	0.965	‘bior3.1’
	2	0.086048	‘bior1.1’	1.204673	‘bior1.1’	0.951	‘rbio3.1’
	3	0.086048	‘rbio1.1’	1.204673	‘rbio1.1’	0.878	‘bior3.3’
	4	0.121297	‘fk4’	1.698156	‘fk4’	0.877	‘rbio3.3’
	5	0.127758	‘mb4.2’	2.044125	‘mb4.2’	0.868	‘fk4’
**2D DCT**		**0.13654**	**‘DCT’**	**0.955781**	**‘DCT’**	**0.8567**	**‘DCT’**
**3D DWT**	1	0.0298	‘mb4.2’	0.357712	‘mb4.2’	0.977	‘bior3.1’
	2	0.0299	‘db2’	0.358703	‘db2’	0.977	‘rbio3.1’
	3	0.0299	‘sym2’	0.358703	‘sym2’	0.905	‘rbio3.3’
	4	0.0304	‘bior1.1’	0.419103	‘fk4’	0.895	‘fk4’
	5	0.0304	‘db1’	0.425066	‘bior1.1’	0.891	‘bior3.3’
**3D DCT**		**0.0193**	**‘DCT’**	**0.232131**	**‘DCT’**	**0.915**	**‘DCT’**

## Data Availability

All data and code can be found at the following link: https://github.com/aslepyan/TactileWavelets, accessed on 25 June 2024.

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
