# Peer review of "Wavelet Transforms Significantly Sparsify and Compress Tactile Interactions"

_sensors, 2024, doi:10.3390/s24134243_

Round 1
Reviewer 1 Report
Comments and Suggestions for Authors
In the work, a library of 12 orthogonal and biorthogonal wavelet transforms as sparsifying transforms for tactile interactions 13 were constructed and compared their tradeoffs in compression and sparsity. The results may be beneficial for free up computational resources for real-time processing on computationally 23 constrained mobile platforms like neuroprosthetics. Some points are better addressed before further evaluatation.
1) High resolutions figures should be provided in Fig. 2
2)How the size change of the sparsest wavelets depending on the reconstruction error?
3) For the discussion part, in order to compare the mechanism and deep analysis, the work: doi.org/10.1016/j.sna.2024.115275 can be compared.
4) The reason for 2D spatial 357 transforms retained spatial structure with less error deviation shoud be given.
Reviewer 2 Report
Comments and Suggestions for Authors
1. The introduction should more clearly outline the gap in the current research and how this paper aims to fill it. Provide more detailed background information on wavelet transforms and explain their suitability compared to other transforms like DCT. Emphasize the novelty of the research more explicitly. It is recommended that the authors consider incorporating relevant content in this regard to enhance readers' understanding of the issues within this field. This addition would also serve as a valuable reference for readers. Relevant literature includes:
2016 International Conference on Manipulation, Automation and Robotics at Small Scales (MARSS), Paris, France, 2016, pp. 1-4, DOI: 10.1109/MARSS.2016.7561748.
2.Simplify complex sentences as much as possible. Break long sentences into shorter ones to improve readability.
3.The methods section should provide a more detailed description of the experimental setup, including dataset preprocessing, parameters of the wavelet transform, and the software/hardware used. For example, provide more information about the dataset, including how the data was collected, the nature of the 26 objects, and any preprocessing steps.
4.Include more comparative charts to show the performance differences between various wavelet transforms and the Discrete Cosine Transform (DCT), highlighting key findings.
5.Explain the rationale behind choosing specific wavelet families and the importance of their properties (such as vanishing moments and symmetry).
6.Ensure tables are easy to interpret, with clear headings and units. Highlight key performance metrics in the tables (e.g., use bold or different colors to mark the best-performing wavelets).
7.Ensure all citations are correctly formatted and consistently styled according to the journal's guidelines.
Round 2
Reviewer 2 Report
Comments and Suggestions for Authors
comments well addressed